# Soil Thermophiles and Their Extracellular Enzymes: A Set of Capabilities Able to Provide Significant Services and Risks

**DOI:** 10.3390/microorganisms11071650

**Published:** 2023-06-24

**Authors:** Juan M. Gonzalez, Margarida M. Santana, Enrique J. Gomez, José A. Delgado

**Affiliations:** 1Institute of Natural Resources and Agrobiology, IRNAS-CSIC, Avda. Reina Mercedes 10, E-41012 Sevilla, Spain; engomez@ucm.es; 2Centre for Ecology, Evolution and Environmental Changes (cE3c) & Global Change and Sustainability Institute (CHANGE), Faculdade de Ciências da Universidade de Lisboa, 1749-016 Lisboa, Portugal; mmcsantana@fc.ul.pt; 3Department of Engineering, University of Loyola, Avda. de las Universidades, E-41704 Dos Hermanas, Spain; jadelgado@uloyola.es

**Keywords:** thermophiles, soil, extracellular enzyme activity, enzyme persistence, climate warming, drought, temperature, organic matter, nutrient cycling, aridity

## Abstract

During this century, a number of reports have described the potential roles of thermophiles in the upper soil layers during high-temperature periods. This study evaluates the capabilities of these microorganisms and proposes some potential consequences and risks associated with the activity of soil thermophiles. They are active in organic matter mineralization, releasing inorganic nutrients (C, S, N, P) that otherwise remain trapped in the organic complexity of soil. To process complex organic compounds in soils, these thermophiles require extracellular enzymes to break down large polymers into simple compounds, which can be incorporated into the cells and processed. Soil thermophiles are able to adapt their extracellular enzyme activities to environmental conditions. These enzymes can present optimum activity under high temperatures and reduced water content. Consequently, these microorganisms have been shown to actively process and decompose substances (including pollutants) under extreme conditions (i.e., desiccation and heat) in soils. While nutrient cycling is a highly beneficial process to maintain soil service quality, progressive warming can lead to excessive activity of soil thermophiles and their extracellular enzymes. If this activity is too high, it may lead to reduction in soil organic matter, nutrient impoverishment and to an increased risk of aridity. This is a clear example of a potential effect of future predicted climate warming directly caused by soil microorganisms with major consequences for our understanding of ecosystem functioning, soil health and the risk of soil aridity.

## 1. Introduction

Soil health and function are strictly linked to microbial activity [1,2,3,4,5]. A large number of processes are carried out mainly or exclusively by microorganisms, and this broad range of activities represents a major asset for soil maintenance and response to multiple variables leading to changing conditions. One of the main factors influencing the functional redundancy of soil processes is microbial diversity [1,5,6,7,8]. Soils are highly heterogeneous, and they present a huge microbial diversity and abundance [9,10]. Current estimates suggest that 1 g of soil contains about 10^10^ prokaryotic cells and includes about 30,000 different microorganisms [7,9,11]. The duplicity of metabolic capabilities allows soils to preserve functionality, maintaining a stable environment, in spite of drastic changes. Otherwise, a significant decrease in microbial diversity would represent a serious handicap on soils being able to maintain current balances and activities, which would negatively affect soil health and productivity [5,7].

Within the research on the almost unmeasurable high microbial diversity existing in soils, most work has been carried out on their major components, while the low-abundance microorganisms have been poorly considered. This represents a significant limitation because a large number of microbial processes with high relevance to, for instance, the biogeochemical cycling of elements are performed by minority microorganisms. This is, for example, the case for ammonium oxidation, denitrification, metal reduction/oxidation, sulfur oxidation, sulfate reduction, methanogenesis and the decomposition of specific recalcitrant pollutants; these processes are generally carried out by groups represented within the minorities of the natural microbial communities [12].

Within the vast microbial diversity of soils, a permanent component consists in thermophilic bacteria. Although thermophiles are expected to inhabit high-temperature environments, such as hot springs, geothermal areas or compost piles, different reports have confirmed the ubiquitous presence of thermophiles in all examined soils from a wide range of latitudes [13,14,15,16,17,18]. Cold and temperate soils hold thermophiles including distinctive taxa, but the major representatives are *Geobacillus*-related genera [13,14,15,16,18]. This study will focus on the potential role and consequences of these soil thermophiles within a perspective of global climate warming.

A question has arose about the timing available for thermophiles to grow, assuming these microorganisms are inhabiting the upper soil layers of cold or temperate environments. Previously, an analysis of average number of hot days against latitude [15,18,19] suggested the occurrence of a significant number of hot days (e.g., above 100 hot days/year, around 37° N, in Seville, Spain) [18,19] when soil thermophiles would have an opportunity to grow and show significant activity. At higher latitudes (i.e., above 50° N), the number of hot days is generally low (e.g., around 1–2 hot days/year, around 52° N, at Cambridge, UK) [18,19], but this could be enough to provide time for maintaining thermophile populations and a minimum extracellular enzyme stock in the soil environment. These soil thermophiles survive well through low-temperature periods [20].

Another group of microorganisms that must be mentioned when considering microorganisms thriving under periods of increased temperatures are the thermophilic fungi [21] also present in soils. The role of these fungi in soils during high-temperature events has not been clearly defined yet [21], and additional research is required. Nevertheless, thermophilic fungi can have a role in compost piles, where high temperatures are maintained for much longer time periods than in upper soil layers. Compost piles typically contain a high organic matter load and generate, as a result of microbial growth, a high-temperature environment [22,23,24,25].

Soil organic matter is a major reservoir of C with the potential to greatly influence global climate [3,4]. Most organic carbon is present in the upper soil layers [26]. Rich soils contain a high content of organic matter, represented by a variety of complex compounds. Soil organic matter, besides its carbon content, also includes other elements, some of them critical for plant and microbial growth, such as nitrogen, sulfur and phosphorous, often required as major soil fertilizers [27,28]. Furthermore, complex organics such as humic acids can complex with those elements as well as with metals [29,30]. Within the soil organic matter, large polymers and humic acids need to be broken into smaller compounds or monomers for microorganisms to be able to be taken up and processed as sources of energy and/or biomass [31,32,33]. This breaking down of complex compounds into smaller ones is mediated by extracellular enzymes. In fact, the bottle-neck for soil organic matter mineralization is this step involving the extracellular enzymes [3,19,34]. Microbial extracellular enzyme activity is highly related to organic matter mineralization in soils and has been proposed as a major indicator to evaluate the sink-link issue with the soil–atmosphere C balance [3,4], a critical parameter for modeling climate predictions [5,35,36,37,38].

Extracellular enzyme activity has been proposed as an indicator of soil microbial activity [39,40,41], and it is commonly measured in ecological studies [37,40,41,42,43,44]. Soil thermophiles have been reported as a major source for extracellular enzymes dominating the pool of enzymes in soils [19] because they present higher total activity than the corresponding enzymes from mesophiles (Figure 1) [18,19,45]. Thus, high-temperature events are expected to enhance soil organic matter processing due to the activation of extracellular enzymes from thermophiles. In addition, the current scenario of global warming suggests an expected increase in frequency and duration of high-temperature events in the coming years [4,5,7,46]. As well, high temperature in soil upper layers implies an increased evaporation and therefore a decrease in water content in soils, leading to increased desiccation. Recent reports have also proposed that some soil microorganisms (including some thermophiles) have adapted to dryness by developing extracellular enzymes and metabolisms able to work optimally under dry conditions (at water activity, a_w_, between 0.3 and 0.8) [43,45]. The levels of desiccation showing maximum extracellular activity by soil thermophilic xerophiles can occur at values below the reported limit for microbial growth (a_w_ 0.605) [47]. Besides extracellular enzyme activity by thermophiles, these cells have been reported to actively decompose recalcitrant pollutants at high temperature [17,48,49,50,51] and under dry conditions [43,45,48], suggesting that cell activity is significant under those extreme conditions in soil upper layers, where these microorganisms can be potential important bioremediation agents [17,48,49,50,51,52,53].

Current modeling of global warming shows important limitations on the incorporation of microbial parameters into model predictions [3,53,54,55]. Under these circumstances of novel microbial activities and microorganisms with previously under-studied capabilities, current previsions from global warming perspectives should be re-examined by incorporating these novel views and the current knowledge on the direct influence of previously underestimated microorganisms. This study will present some potential positive and negative consequences of the activity of soil thermophiles from the perspective of maintaining soil health and productivity.

## 2. Singularity of Soil Thermophiles and Thermophilic Extracellular Enzymes

Soil thermophiles represent a singularity in the microbial communities from cold and temperate soils. However, thermophiles are ubiquitous inhabitants of soils, and their presence as viable cells with an important role and capabilities to survive under those conditions have been reported [14,15]. Thus, thermophiles thriving in temperate environments, depending on periodic/sporadic high-temperature events, can show some growth and produce extracellular enzymes that are required to process organic matter [18,19]. These extracellular enzymes might persist in the environment [56] and become active under heat events. Consequently, these extracellular enzymes actively participate in soil organic matter decomposition [19,43,56].

Mesophilic microorganisms present extracellular enzymes with optimum activity at moderate temperatures (i.e., generally measured around 30 °C or below) [57,58,59], but the extracellular enzymes from thermophiles present optimum activity at temperatures above 50 °C [19,45]. Thus, the activity by extracellular enzymes from thermophiles can be easily detected by carrying out enzyme assays at high temperature (50 °C to 70 °C), and therefore, mesophilic and thermophilic activities can be differentiated. Results discriminating the activity over a range of temperatures (from 5 °C to 95 °C) in a variety of soil samples showed unexpected results [19]. The results clearly indicated that thermophilic activities were always higher than mesophilic ones in all soils tested (Figure 1) [18,19]. This suggests that thermophilic extracellular enzyme activities are dominant in soils. It is important to note that the pioneering work [19] and some subsequent studies [43,45] included samples from soils exposed to hot temperatures and others from cold environments. Additional studies have shown significant roles of thermophiles at higher latitudes [16,17,49,50,51,52], corroborating that soil thermophiles can also show significant environmental activity in relatively cold climate zones.

Because soil thermophiles represent a minority fraction of the total microbial community in cold and temperate soils [13,15,18,43], it is required to look for different potential scenarios to explain that large activity measured in the thermophile temperature range. An easy explanation would be that soil thermophiles show a very high production of extracellular enzymes and/or these enzymes present higher activity than their mesophilic counterparts. Although enzymes from thermophiles have been reported to present higher activity than those from mesophiles [60,61], the difference is not likely to be able to explain the much higher total activity measured in soils due to thermophiles (a minority group) than due to the total mesophilic microbial community, which presents a much higher abundance. According to previous estimates [15], the fraction of thermophiles in soils is generally below 1% of the total community. A potential justification for that large activity at high temperatures in soils could be that soil thermophiles could produce a large amount of extracellular enzymes during hot periods or extreme heat events. Soil thermophiles, such as *Geobacillus* related taxa, require extracellular enzymes to access complex organic matter in soils and grow, so a high production of enzymes is needed for their development. Nevertheless, the relatively low abundance of thermophiles suggests that the production of thermophilic extracellular enzymes could not be as high as needed—in relationship to the production by mesophiles showing equivalent metabolism—to explain the higher activity in the thermophile temperature range. The level of extracellular enzyme production that could be potentially needed to explain that scenario is likely to be out of reach for the soil thermophilic cells. Otherwise, during hot periods, thermophiles could respond to heat events by growing and producing a moderate amount of enzymes that would persist in the environment over time. This could result in a progressive accumulation of thermophilic enzymes in the environment. These enzymes should be able to persist in the environment at least until the next hot event, and, at that moment, the thermophilic enzymes will show their full activity. For this to be a reasonable explanation, the thermophilic extracellular enzymes should show longer persistence in the soil environment. The potential for accumulation of extracellular enzymes in soils could represent a singular strategy that would allow thermophilic cells to start growing rapidly when the right growth conditions arrive in the ecosystem (e.g., an extreme heat event or hot days during the summer period). A rapid response would take advantage of even the shortest periods at high temperature to recover and grow. During these hot periods, those thermophilic enzymes could decompose soil organic matter, allowing a variety of organisms (both microorganisms and plants) to profit from that release of smaller compounds (and monomers) readily available as substrates for growth and energy [62]. A mechanism facilitating a rapid growth response to high-temperature events (i.e., extracellular enzyme accumulation) represents an interesting adaptive feature for thermophiles to thrive in cold and temperate environments. This type of strategy has not been reported before for microorganisms.

For extracellular enzymes to accumulate in the environment, two aspects need to be fulfilled: a relatively long production of extracellular enzymes and a long persistence in the environment. Generally, enzymes from thermophiles (as well as from other extremophiles) have been reported to present higher durability than those from mesophiles, resulting from a higher stability and resistance to external factors (detergents, denaturants, decomposition, etc.) [60]. Assuming enzymes are produced at a relatively high rate and then show a long persistence in the environment, they could progressively accumulate in soils [56]. In this way, thermophilic extracellular enzymes could generate an active enzymatic pool in soils, readily available to catalyze complex organic matter decomposition as soon as temperatures rise. A recent report [56] has shown that extracellular enzymes from thermophiles are able to persist for a longer time in soils than those from mesophiles (Figure 2). Thermophilic extracellular enzymes maintain their activity in soils even at the highest temperatures and desiccation levels reached in the upper soil layers (Figure 2). Mesophilic enzymes are rapidly denatured during extreme heat events, including summer periods, and their persistence is lower than that for soil thermophiles [56,63]. Thus, the extracellular enzymes from thermophiles are able to persist in the environment, representing a soil asset that could allow the rapid growth of microorganisms, both mesophiles and thermophiles, and so promote soil health and functioning. This unique strategy proposed for soil thermophiles represents a singular mechanism to survive in environments (i.e., cold and temperate soils) considered adverse for thermophiles.

High temperature of upper soil layers implies increased evaporation, leading to a reduction in water content and desiccation. It has been generally thought that dried soils present poor or near-null biological activity. Nevertheless, recent work has shown that specific microorganisms present optimum extracellular enzyme activity under dry conditions [43,45,64,65]. This is the case for some soil thermophiles, among other mesophilic bacteria (i.e., *Deinococcus*). *Deinococcus radiodurans*, a wide spread soil bacterial species, has been reported to be a model microorganism for resistance to desiccation [66], and some of its extracellular enzymes can present optimum (maximum) activity under dried soil conditions (a_w_ 0.40–0.55) (Figure 3) [45]. Soil thermophiles, those inhabitants of soils frequently exposed to high temperatures and droughts, exhibit an interesting feature: they present optimum extracellular enzyme activities at very low water content (a_w_ 0.3–0.7). Some thermophilic enzymes from sites exposed to hot climate have been shown to reach optimum activities under dry conditions (a_w_ < 0.7), but thermophiles from cooler locations present optima under wet conditions (a_w_ > 0.9). Most mesophilic enzymes in those natural soil samples always presented optimum values in aqueous solutions (a_w_ > 0.9). These results showed that soil thermophiles can adapt to thrive in a variety of environments and have the capacity to adapt their enzyme activities to extremely dry conditions [43,45].

Furthermore, these thermophiles survive and remain viable under these dry conditions (a_w_ 0.5), as shown by a comparative study on pollutant decomposition by *Geobacillus* (showing maximum pollutant decomposition at a_w_ 0.5) compared to *Rhodococcus* spp. (showing maximum decomposition at a_w_ > 0.9) [48]. This qualifies soil thermophiles from hot and dry environments as some of the most xerophilic cells reported on Earth. So far, the lowest water activity allowing growth is 0.6 for the fungus *Xeromyces bisporus* [47], and most microorganisms do not show growth below a_w_ 0.8 [67]. Some soil thermophiles can show optimum extracellular enzymatic activity [43,45] and ability to decompose pollutants [48] at a_w_ around 0.5. Consequently, singular features of soil thermophiles include their potential adaptability to extreme drought and high temperatures, which are valued because of their consequences for the environment [43,45,56], providing evidence of optimal activity under those extremes (Figure 3), as well as a great potential for biotechnological applications in high-temperature and non-aqueous treatments and processes [68,69,70,71].

## 3. Organic Matter Decomposition

Soil organic matter is largely processed by microorganisms and dependent on concentration, composition and a variety of biotic and abiotic factors [3,35,37,40,41,59]. Most research has been performed to understand microbial organic matter decomposition in soils during standard conditions (temperature and high-water-content conditions) [3,35,37,40,41,59] and after wetting events [72,73,74,75,76], but scarce information is available on the potential of microbial activity under extreme events (i.e., high temperature, desiccation) [15,19,43,77]. Soil thermophiles are able to thrive under periodic or sporadic hot and dry periods (and survive in the cold), unlike what is assumed for most microorganisms. The relevance of thermophiles in soil ecosystems during those extreme events must be evaluated so that researchers and technologists can incorporate the information into current local and global system management, models and predictions.

Microbial organic C processing in soils was thought to be inhibited during extreme events (i.e., high temperature and desiccation) [57,58,78,79,80]. In the last decade, researchers have started to understand that some microorganisms are able to occupy niches that are not able to support growth of others, leading to alternating feast–famine cycles in the environment [81,82,83]. As an example, different microorganisms packed tightly along a temperature gradient [84] and thedistinctive taxa naturally distributed according to the environmental conditions and outcomes of competition with other microorganisms. Soil thermophiles have been confirmed [13,15,18,43] to present optimum activity under extreme conditions in which other microorganisms do not grow. During the temperate and cold periods, those soil thermophiles persist under near-zero or maintenance growth stages [14,62,85]. In fact, periods of growth limitation and even inhibition have been reported to be the most common conditions for most microorganisms in nature [86,87]. Thus, growth limitation and the wait for appropriate conditions to arrive should be considered as general features for microorganisms in the environment and, specifically, a way of living for soil thermophiles [14,19].

Organic matter is a parameter of major importance to determine soil health and productivity [88,89]. Microorganisms are major participants in maintaining soil functioning [90,91,92], including the working of biogeochemical cycles of elements sustaining plant growth and interactions with plants [27,90,92,93,94]. During extreme events (i.e., extreme temperature and desiccation), soil thermophiles can perform some of these functions during periods when other microorganisms are inhibited. Soil thermophiles (as mentioned above) utilize extracellular enzymes to start the decomposition of complex organic matter in soils. This first step is generally the limiting bottle-neck for soil organic matter mineralization [3,19,34]. The decomposition of organic matter into smaller compounds or monomers that can directly be incorporated by cells fosters microbial growth. During growth, aerobic respiration generates CO_2_ as a result of complete organic matter mineralization [32], and the metabolism of those organic molecules leads to the release of other inorganic compounds (i.e., ammonium, sulfate, phosphates) [15,32,85,93]. Thus, the role of soil thermophiles, both those thriving at high temperatures and those adapted to thrive under dry conditions, should be accounted for when evaluating microbial activity and the consequences on local and global scales, in order to fully understand ecosystem functioning and the fate of C at the soil–atmosphere equilibrium of relevance and thereby achieve reasonable predictions on climate impact [4,5].

Soil organic matter, including humic acids, presents a significant content of nitrogen [29,30,95,96,97,98]. Proteinaceous compounds contain nitrogen, and the consumption of these compounds represents an important source of N for microorganisms. Excess N from this metabolism can be released into the environment, generally in the form of ammonium, and it can be a substrate for other bacteria directly involved in critical steps of the biogeochemical cycle of N (e.g., anammox, nitrifiers, denitrifiers) [99]. The inorganic N released, with synthesis of ammonium, nitrate and nitrite, represents a significant source of N for all soil organisms, specifically plants. Inorganic N is frequently a growth-limiting nutrient for plants [100]. Soil thermophiles have been cited to release N as ammonium at a rate similar to that reported for super-ammonifiers in the mesophile temperature range. Thus, this source of inorganic N can be maintained and potentially enhanced during extreme heat and dryness periods by the role of soil thermophiles and their decomposition of soil organic matter.

Processing of soil organic matter, besides C and N, affects other elements. One of the elements most relevant for plant and microbial life is sulfur. S is often required in plant cultures as a fertilizer because the availability of S in soils is frequently low. In this respect, most S in soils is part of soil organic matter (about 90% of total soil S) [101,102,103], for example, integrated in humic acids and proteinaceous compounds [29,30]. Nevertheless, the capacity of most soil bacteria (analyzed at 20 °C, in the mesophile temperature range) to mineralize organic S has been shown to be highly limited [101,102] and has been suggested to be frequently unable to support the S requirements for soil organisms (microorganisms and plants). Soil thermophiles, during extreme heat and desiccation periods, can process soil organic matter releasing inorganic S, mainly as sulfate [15,85,104]. A significant release of sulfate has been reported for soil thermophiles at a much higher rate than that by soil mesophiles [15]. A putative pathway involved in the release of inorganic S (i.e., sulphate) has been described [85,104], supporting the potential for S cycling in soils carried out mostly by thermophiles within the phylum Firmicutes, bacteria closely related to the *Geobacillus* genus. These soil thermophiles could reduce S limitation in soils during high-temperature and dry periods by increasing the recycling rate of organic S [85,104].

Another major microbial growth-limiting nutrient is P. Phosphates are relatively abundant in soils, but they are generally complexed into organic matter and minerals [105,106,107,108,109,110]. Numerous microorganisms are able to release phosphates, making P freely available for biological utilization and incorporation into the cell metabolism of both microorganisms and plants [32,107]. The role of phosphatases is important for the solubilization of phosphates in soils [32,43,107]. Some soil thermophiles (i.e., *Geobacillus* related strains) have been shown to solubilize phosphates during growth, which represents an additional feature of relevance for the working of soil ecosystems during high-temperature and desiccation periods.

Soil thermophiles metabolism, as shown above, is directly implicated in a number of features related to the cycling and processing of organic matter. The role of their extracellular enzymes is critical and limits the rate of the subsequent pathways and metabolism performed by the thermophilic cells. An additional aspect of interest is the capability of soil thermophiles to decompose recalcitrant pollutants under extreme conditions. This has been shown by various research teams. An example is the ability of microorganisms in natural soil samples and of *Geobacillus* strains isolated from soils to decompose halocompounds (e.g., chlorophenol) under high-temperature and low-water-activity conditions [48]. Another example was reported of the major role of thermophiles in the decomposition of bituminous hydrocarbons in soils in a relatively cold climatic zone (59 °N) [17]. The degradation of n-hexadecane in soil has been carried out by thermophiles [51]. Another report pointed out the role of thermophiles in crude oil degradation in soils [50]. The decomposition by thermophiles of oily food residues in soils has also been confirmed [49]. Thus, soil thermophiles expand the time when and conditions under which bioremediation could be carried out in soils [17,48,49,50,51,52], including periods of high temperature and drought in upper soil layers.

Herein, the role of thermophiles has been summarized in relationship to natural soil systems. However, similar roles are to be expected if these processes are incorporated into bioreactors [50,111] or composting [112,113,114,115]. Under these technological scenarios, thermophiles could be functioning under most defined and optimum conditions, showing increased relevance to organic matter processing, inorganic nutrient release and pollutant decomposition. Biotechnological processes can significantly profit from the capabilities of soil thermophiles.

## 4. Positive Implications (Services)

The use of soil thermophiles as potential biofertilizers has been suggested and is supported by the evidence shown above. Soil thermophiles could significantly contribute to C processing and the release of inorganic ammonium, sulfate and phosphate, which can be directly utilized by plants [62,85,93]. As a consequence of this active participation of soil thermophiles in nutrient cycling, one could suggest that periods of extreme climate events also contribute to maintaining soil health and productivity, for example, by enriching soil with available inorganic nutrients that can be readily used for plant growth.

Soil thermophiles’ activity during hot and dry periods can represent an additional level of redundancy that enhances the role of microbial diversity by providing pathways leading to nutrient cycling and sustainability during extreme conditions [1,94,104,116,117]. Soil health requires microbial diversity so that changes can be compensated through different alternative pathways and soil functioning can continue providing similar services [6,8,118].

In order to confirm soil thermophiles’ positive effects on plant growth, a series of studies have shown the potential capabilities of those soil thermophiles to enhance plant growth [62,85,93] and drought tolerance [62,94]. These reports showed that plants significantly benefit from soil thermophiles’ activities, which contribute to compensating nutrient deficits by increasing decomposition of soil organic matter and releasing essential plant nutrients.

In addition to the participation of soil thermophiles in the biogeochemical cycling of elements (C, N, S, P), these microorganisms can provide bioremediation services under high-temperature and dry conditions. Previously, the potential bacterial decomposition of pollutants during extreme heat and dry events had been considered to be highly limited [119,120]. Studies on soil thermophiles have shown the importance of soil thermophiles in these processes, contributing significantly to decomposing pollutants under these conditions [17,48,49,50,51]. In fact, high temperature, for example, can contribute to increasing solubility of some pollutants, increasing the accessibility to thermophilic cells [121]. Similarly, drought enhances evaporation, and this can lead to pollutant concentration, which can facilitate decomposition by overcoming affinity issues at low concentrations [122]. Thus, thermophiles offer enhanced potential as soil bioremediation tools [17,48,51].

As mentioned above, soil thermophiles under moderate average conditions thrive in the environment when periodically or occasionally they are able to grow and show significant activity, thus contributing moderately to the sustainability and productivity of the soil ecosystem. Caution must be taken because uncontrolled increased exposure of soils to extreme climate events could lead to differential effects considered below.

## 5. Negative Implications (Risks)

Above, we have presented a number of positive implications of and services potentially provided by soil thermophiles during extreme heat and drought events. However, there are potential negative effects that need to be considered.

Excessive organic matter decomposition can lead to soil impoverishment [123,124]. Although additional research is required, Santana and Gonzalez [18] have suggested a potential correlation between the activity of soil thermophiles and soil organic content throughout Europe. Lower latitudes present higher temperatures, and it is expected that soil thermophiles show higher activity there; Southern Europe generally presents lower soil organic matter content than Northern Europe. Higher latitudes correspond to cooler climatic zones, so soil thermophiles and their extracellular enzymes will show limited activity.

Based on current climate change predictions, raising temperatures will likely result in a higher frequency and duration of extreme temperature events and droughts [3,4,124,125]. If so, the periods and duration of hot days and periods of drought will increase, leading to increased opportunities for growth of soil thermophiles and the enhanced activity of thermophilic extracellular enzymes. This potential increase in thermophilic activity could lead to excessive organic matter decomposition, with two major aspects to be considered.

A potential increased soil organic matter decomposition could lead to soil impoverishment due to a decrease in soil organic matter content. Extreme heat and dry periods could lead to increased activity derived from soil thermophiles. Future climatic scenarios and predictions suggest that the expected conditions will lead to an increase in soil thermophile-derived activities, which could potentially pose a significant risk of progressive soil impoverishment and CO_2_ release to the atmosphere, due to excessive organic matter mineralization. Increased extreme climatic conditions might lead to poor plant health and plant coverage reduction as a result of high temperature and reduced water availability [46,94,119,126,127]. Soils poorly covered with vegetation will be increasingly affected by radiation, heat and desiccation [128,129,130,131]. Thus, increased climate warming would lead to increased soil thermophile activity, which may result in an increased risk of soil aridity.

A second potential aspect that needs to be considered is that the above-mentioned enhanced soil organic matter decomposition should result in the increased release of inorganics (i.e., ammonium, sulfate, phosphate). Because plants under extreme heat and drought conditions might be under severe stress situations, plant uptake of inorganic nutrients might be poor, and so those available inorganic compounds might be easily lost, for example, in runoff water [132,133]. This might enhance nutrient loss and lead to increased risk of soil impoverishment and aridity.

Under the current global climate warming scenario, an increase in the activity of soil thermophiles is expected. If plants’ growth and potential adaptations are not supported by the warmer climatic conditions expected to occur in the future [125,134,135], a reduction in plant coverage of soil surface and an increased activity of soil thermophiles are suggested to result in a progressive soil impoverishment and risk of aridity. A large amount of thermophilic extracellular enzymes accumulated in soils [19,43,45] leads to the hypothesis that soils would experience a reduction in their organic matter content if temperatures and dryness increase through climate warming. These risks should be proportional to climate events, and these effects will be related to latitude and altitude, among other factors influencing environmental conditions in different soil systems [135,136,137]. Although moderate soil thermophile activity can result in positive benefits for soil sustainability and health, an excessive or enhanced activity of soil thermophiles could certainly have strong negative effects on soils. Evaluating these effects and potential consequences requires further investigation.

## 6. Conclusions

Within the huge microbial diversity existing in soils, some minority components could present a significant potential and certainly provide highly relevant value by expressing functional redundancy and warranting soil response to potential changes and drastic events, including climate changes predicted in a near future. In this study, we have focused on one minority group of soil communities, the soil thermophiles, and analyzed potential benefits and risks derived from expected climate changes.

Soil thermophiles can contribute positively to nutrient cycling by processing soil organic matter, including recalcitrant compounds and pollutants, and releasing inorganic nutrients complexed within the organic matter. During hot and dry periods, thermophiles can grow, consuming organic matter and producing extracellular enzymes to decompose organic complexes. Those enzymes can persist in the environment and show activity during hot periods. As a consequence of this organic matter mineralization, inorganic nutrients (ammonium, sulfate and phosphate) will be made available to soil organisms, including plants.

Nevertheless, poorly covered soils can be exposed to increased radiation. Increased soil temperature and droughts as a result of climate change will lead to increased activity of soil thermophiles, which might induce undesirable effects such as nutrient impoverishment (as a result of a decreased reduction in soil organic matter and potential runoff of soluble inorganics) and an increased risk of soil aridity.

At present, current previsions on climate warming barely incorporate direct effects of microbial activity on a global scale. Herein, we provide evidence and information suggesting that potentially significant effects of microorganisms can occur. Additional investigation is required to confirm the likelihood of those services and risks within future climate scenarios. A minority group in the microbial communities, the soil thermophiles, could potentially switch from positive (current/past scenario) to negative (potential future scenario) effects as a result of predicted climate changes. Climate change predictions foresee increased temperature and droughts, which could result in a spiral fostering negative effects and risks, decreasing soil health and productivity.

## Figures and Tables

**Figure 1 microorganisms-11-01650-f001:**
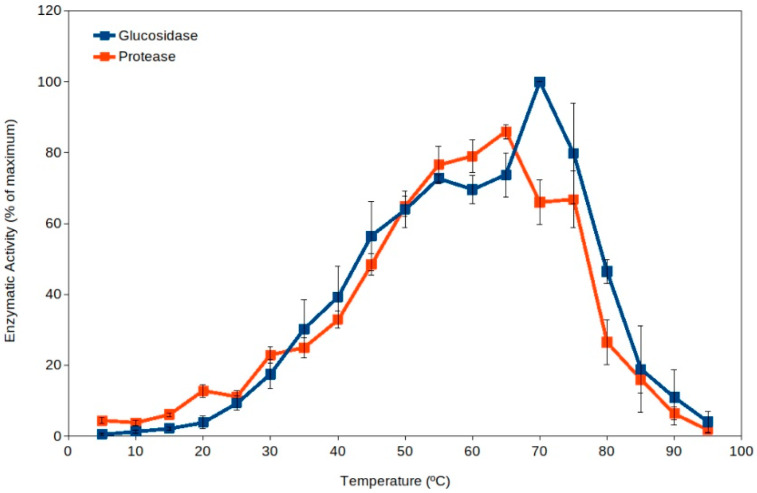
Extracellular enzyme activity in soils over a broad range of temperatures (5 °C to 95 °C). Maximum activity was observed at temperatures in the thermophilic microorganisms range (55 °C–75 °C). Data averaged from Gonzalez et al. [19]. Blue squares, glucosidase; red squares, protease. Error bars, one standard deviation.

**Figure 2 microorganisms-11-01650-f002:**
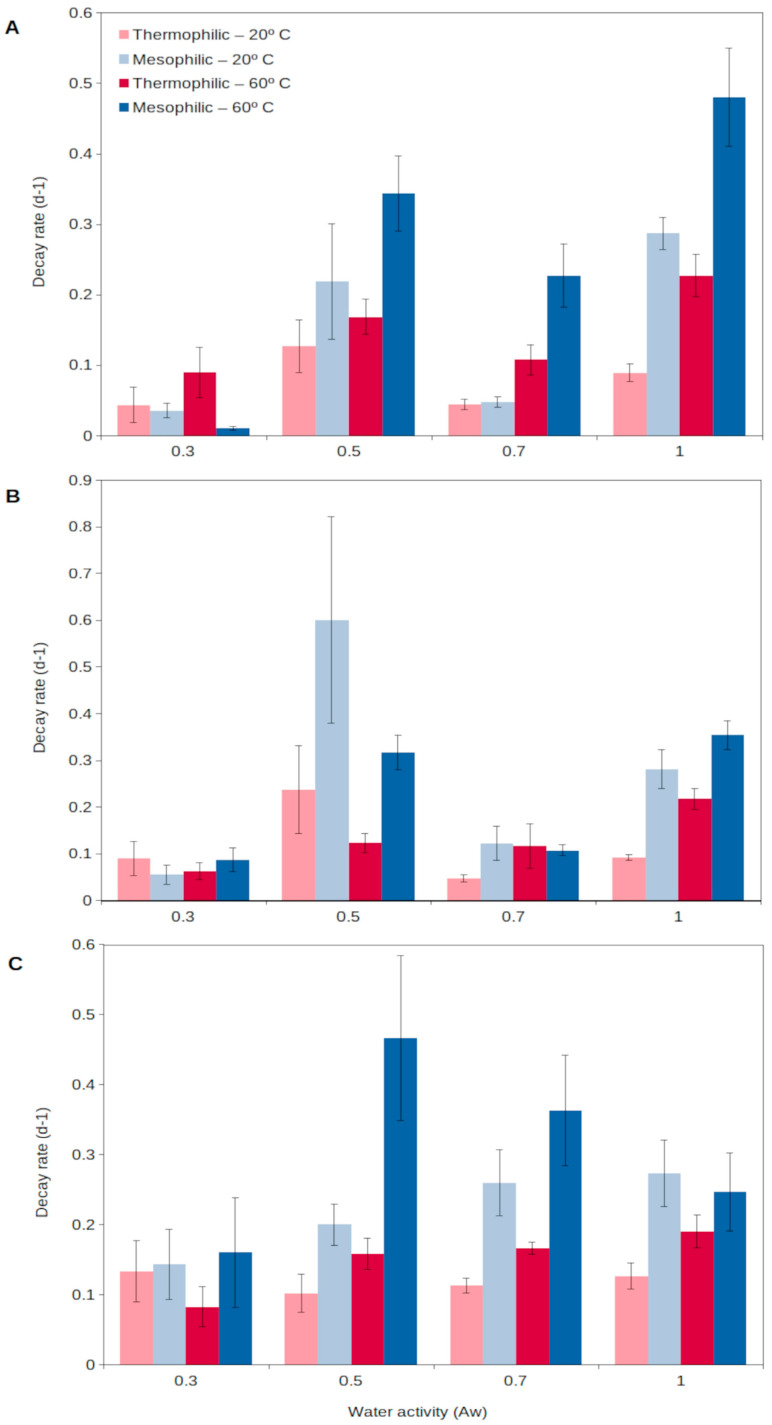
Decay rate of natural extracellular enzymes in soils at different temperatures, 20 °C (light colors) and 60 °C (dark colors), corresponding to the mesophilic (blue) and thermophilic (red) microorganisms, respectively, over a wide spectrum of water contents (water activity, a_w_) for 3 different enzyme activities: (**A**), glucanase; (**B**), phosphatase; (**C**), protease. High decay rate indicates short persistence, and low decay rate shows long persistence. Data averaged from Gomez et al. [56]. Error bars, one standard deviation.

**Figure 3 microorganisms-11-01650-f003:**
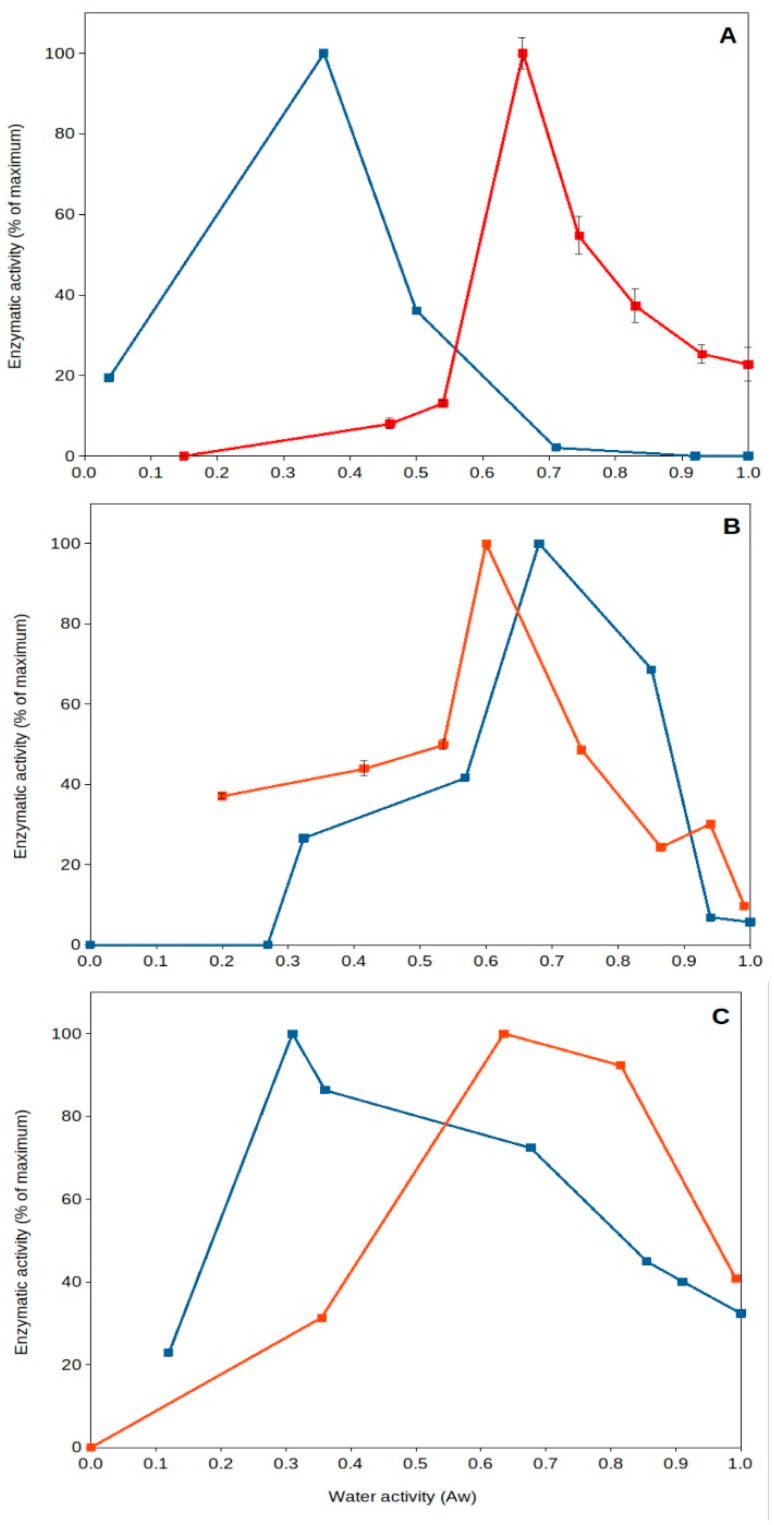
Natural extracellular enzyme activity from Southeastern Spain soils versus water activity. A couple of examples showing optimum extracellular enzyme activity under severe desiccation. Red squares and lines, soil samples collected from Coria del Rio (Seville); blue squares and lines, soil samples collected from Benaocaz (Cadiz). (**A**), glucanase; (**B**), phosphatase; (**C**), protease. Data extracted from Gomez et al. [43]. Error bars, one standard deviation.

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
