# Peer review of "Soil Thermophiles and Their Extracellular Enzymes: A Set of Capabilities Able to Provide Significant Services and Risks"

_microorganisms, 2023, doi:10.3390/microorganisms11071650_

Round 1
Reviewer 1 Report
I would like to express my gratitude and congratulations to the authors for their high-quality review article. I have provided detailed comments and feedback in the attached document.

Author Response
Thank you for your comments.
Figures 2 b and c have been re-scaled on their Y-axis to show the full error bars as suggested.

Reviewer 2 Report
The authors have condensed a large body of relevant literature into this review. The scientific quality of this ms. is very good. Comments are limited to the improvement of the presentation (below).
P. 1
Abstract
"During this century, ....".
"of reports have described the potential...."
Introduction
"Soil health and function....".
P. 2
"a permanent component are thermophilic...."
"A question has arose about...."
"soil thermophiles would have an opportunity...."
"thermophilic fungi can have a role in....".
"Rich soils contain a high....".
"[27, 28]. Also, complex....".
"[37, 40-44]. Soil thermophiles....".
P. 3
"have adapted to dryness by developing....".
"recalcitrant pollutants at high....".
P. 4
"could respond to heat events by growing....".
"This could result in a....".
"start growing rapidly when the right....".
"to thrive in cold and temperate....".
P. 5
"This type of strategies has not been reported....".
"aspects need to be fulfilled: a relatively long production....".
"as soon as temperatures rise."
"enzymes maintain their activity....".
"their persistence is reduced than that....".
"thermophiles are able to persist....".
"asset that could allow the rapid....".
"and so promote soil health....".
P. 6
"[45]. Soil thermophiles, those.... droughts, exhibit an interesting....".
"Some thermophilic enzymes.... reach optimum activities under dry....".
"environments and have the capacity to fit....".
"so far, the lowest water activity allowing growth (above 0.6) was for the fungus....".
"[43, 45] and ability to decompose....".
"applications in high temperature....".
P. 7
"unlike what is assumed for most....".
P. 8
'in soils was thought to be inhibited....".
"rest of microorganisms do not grow.".
"growth limitation, and the wait for appropriate....".
"environment and, specifically, a way....".
"major participants in maintaining....".
"[90, 92], including the....".
"into smaller compounds or monomers....".
"Excess N from this....".
"nutrient for plants [100].".
"and N, affects other elements.".
"the elements most relevant for plant....".
"most S in soils is part of....".
P. 9
"rate than that by soil....".
"carried out mostly by thermopiles within the phylum Firmicutes, bacteria closely related....".
"by increasing the recycling rate....".
"Numerous microorganisms are able to release phosphates, making P freely available for biological....".
"have been shown to solubilize phosphates....".
"in crude oil degradation....".
"[50, 111] or composting.".
"profit from the capabilities....".
"suggested and is supported by the evidence....".
P. 10
"In order to confirm....".
"dry events had been considered....".
"differential effects considered below.".
"research is required, Santana....".
"temperatures and it is.... show higher activity there. Southern Europe....".
"climatic zones, so soli thermophiles....".
"will increase leading to....".
"impoverishment due to a....".
P. 11
"thermophiles may result".
"A large amount.... leads to the hypothesis....".
"could certainly have strong....".
"consequences requires further investigation."
"risks derived from expected....".
References
correct capitalization [7, 90].
Paper title [8].
Author Response
Thanks. We appreciate your comments, suggestions and detailed reading. Every one of your suggestions have been incorporated in the corrected version of the ms. See the version with track changes to see all these corrections.
